# Optimising Meropenem and Piperacillin Dosing in Patients Undergoing Extracorporeal Membrane Oxygenation Without Renal Dysfunction (MEPIMEX)

**DOI:** 10.3390/antibiotics14090939

**Published:** 2025-09-17

**Authors:** Mar Ronda, M Paz Fuset, Erika Esteve-Pitarch, Josep Llop, Victor Daniel Gumucio-Sanguino, Evelyn Shaw, Daniel Marco Mula, Kristel Maisterra-Santos, Joan Sabater, Xose L. Pérez, Sara Cobo-Sacristan, Raül Rigo, Fe Tubau, Jordi Carratalà, Helena Colom-Codina, Ariadna Padullés

**Affiliations:** 1Department of Infectious Diseases, Hospital Universitari de Bellvitge-IDIBELL, Hospitalet de Llobregat, 08907 Barcelona, Spain; maronse1@gmail.com (M.R.); evelyn.shaw@bellvitgehospital.cat (E.S.); jcarratala@bellvitgehospital.cat (J.C.); 2Department of Intensive Care Medicine, Hospital Universitari de Bellvitge-IDIBELL, Hospitalet de Llobregat, 08907 Barcelona, Spain; mfuset@bellvitgehospital.cat (M.P.F.); vgumucio@bellvitgehospital.cat (V.D.G.-S.); dmarcom@bellvitgehospital.cat (D.M.M.); kmaisterra@bellvitgehospital.cat (K.M.-S.); jsabater@bellvitgehospital.cat (J.S.); josep@bellvitgehospital.cat (X.L.P.); 3Department of Pharmacy, Hospital Universitari de Bellvitge-IDIBELL, Hospitalet de Llobregat, 08907 Barcelona, Spain; eestevepitarch@gmail.com (E.E.-P.); jmllopt@gmail.com (J.L.); scobo@bellvitgehospital.cat (S.C.-S.); 4Centro de Investigación Biomédica en Red de Enfermedades Infecciosas (CIBERINFEC), Instituto de Salud Carlos III, 28029 Madrid, Spain; 5Department of Clinical Laboratory, Hospital Universitari de Bellvitge-IDIBELL, Hospitalet de Llobregat, 08907 Barcelona, Spain; raulr@bellvitgehospital.cat; 6Department of Microbiology, Hospital Universitari de Bellvitge-IDIBELL, L’Hospitalet de Llobregat, 08907 Barcelona, Spain; f.tubau@bellvitgehospital.cat; 7Centro de Investigación Biomédica en Red de Enfermedades Respiratorias (CIBERES), Instituto de Salud Carlos III, 28029 Madrid, Spain; 8Department of Clinical Sciences, Faculty of Medicine and Health Sciences, University of Barcelona, 08907 Barcelona, Spain; 9Pharmacy and Pharmaceutical Technology and Physical Chemistry Department, Faculty of Barcelona, University of Barcelona, 08028 Barcelona, Spain; helena.colom@ub.edu

**Keywords:** extracorporeal membrane oxygenation, therapeutic drug monitoring, critically ill patients, meropenem, piperacillin

## Abstract

**Background/Objectives**: Antibiotic pharmacokinetics (PK) and pharmacodynamics (PD) are altered during extracorporeal membrane oxygenation (ECMO). Meropenem and piperacillin are among the most commonly prescribed antibiotics for infections in this population. However, guidance on dosage adjustments in the ECMO setting remains limited. We aim to assess differences in meropenem and piperacillin concentrations achieved and identify the clinical, physiological, and mechanical factors influencing antibiotic exposure. **Methods**: This is a retrospective, single-centre, observational study comparing an ECMO cohort with a population control group from a prior study, without renal dysfunction. Demographic, clinical, PK/PD parameters, and ECMO-related data were analysed using univariate and generalised estimating equations. For both antimicrobials, the PK/PD target was set at 100%*f*T_>4xMIC_. **Results**: A total of 130 critically ill patients were included: 18 in the ECMO group and 112 in the control group. The mean age was 65 years (23), 67% were male and 26.9% were classified as obese. For meropenem, renal function and ECMO support significantly influenced drug exposure, with PK/PD targets being achieved in 67% of measurements; in contrast, piperacillin exposure exhibited greater variability, primarily driven by renal function and mechanical ventilation. Notably, PK/PD targets for piperacillin were met in only 20% of measurements. **Conclusions**: Our findings highlight the considerable variability in β-lactam exposures and PK/PD target attainment in critically ill patients. This study underscores the importance of therapeutic drug monitoring and individualised dosing in attempts to improve antimicrobial efficacy and patient outcomes in this challenging setting.

## 1. Introduction

The rising prevalence of multidrug-resistant (MDR) microorganisms, coupled with the ever-slower development of new antimicrobial agents, has intensified the need to implement strategies that extend the lifespan of existing antibiotics [1]. One promising approach is the optimisation of antimicrobial pharmacokinetics (PK) and pharmacodynamics (PD) to enhance the probability of therapeutic success [2].

For time-dependent antibiotics such as β-lactams, bactericidal activity is closely linked to the percentage of time of the dosing interval (T) during which the free (*f*) steady-state serum concentration (*f*C_ss_) remains above the minimum inhibitory concentration (MIC) of the causative pathogen (%*f*T_>MIC_) [3,4,5]. To achieve optimal outcomes and overcome resistance, current evidence suggests that *f*C_ss_ should be above 4 to 8 times the MIC throughout the entire dosing interval (100%*f*T_>4-8xMIC_) [6,7,8,9].

Critically ill patients present a unique challenge due to the profound physiological disruptions that significantly alter antibiotic PK parameters [10,11,12,13,14,15]. In the DALI study, one-fifth and 40% of the critically ill patients receiving β-lactam therapy failed to achieve 50%*f*T_>MIC_ and 100%*f*T_>MIC_, respectively [16]. One strategy to overcome suboptimal target attainment with β-lactams is the use of high-dose and prolonged or continuous infusion therapies, which maintain sustained drug concentrations throughout the dosing interval and maximise antibiotic efficacy [17,18,19]. However, even with this dosing strategy, not all patients achieve sufficient antibiotic concentrations [20]. An additional intervention to optimise PK/PD target attainment of β-lactams is therapeutic drug monitoring (TDM) with individualised dose adjustment, which has emerged as a key tool in this area [21]. Some experts advocate systematic TDM and individualised dose adjustments in critically ill patients, a policy supported by recently published TDM algorithms [22,23]. Others suggest a more selective approach, targeting specific subsets of patients likely to benefit from β-lactam TDM [21,24].

The use of extracorporeal membrane oxygenation (ECMO) further complicates this scenario. Widely applied as a rescue therapy for severe acute respiratory distress syndrome and circulatory failure, ECMO introduces additional PK variability through circuit-related absorption and sequestration of antimicrobials [23,25]. ECMO is also associated with a high incidence of nosocomial infections [10,25]. Respiratory tract infections caused by Gram-negative microorganisms are particularly common, often treated empirically with antipseudomonal agents such as meropenem and piperacillin [26,27]. These factors heighten the risk of therapeutic failure, toxicity, and the emergence of resistance, especially when standard antibiotic dosages are used [2,28]. However, data available for guiding the dosing of these antibiotics in ECMO patients are limited [20,29,30,31,32]. The ASAP study described the PK of 11 antimicrobials in intensive care unit (ICU) patients receiving ECMO. Only 56.5% of the concentration profiles achieved the predefined PK/PD target, and those under meropenem or piperacillin, administered by intermittent infusion, exhibited substantial variability in all PK parameters [33]. Studies of antibiotics administered by continuous infusion in ECMO are scarce and often lack a control group, thereby hindering the identification of specific effects of ECMO on drug disposition [20,25,29]. Consequently, robust evidence-based dosing recommendations for this complex population remain scarce.

This study aims to bridge this knowledge gap by comparing meropenem and piperacillin *f*C_ss_ during continuous infusion in critically ill patients with and without ECMO support. By identifying the clinical, physiological, and mechanical factors influencing antibiotic exposure, we seek to generate evidence that supports individualised dosing strategies in order to optimise patient outcomes in this challenging setting.

## 2. Results

A total of 130 patients were included in the study, with a mean age of 65 years (23 years old). The ECMO cohort comprised 18 patients, and the historical control cohort 112. Eighty-seven patients (67%) were male, and fifty-five (26.9%) were classified as obese. The median estimated glomerular filtration rate (eGFR) was 98.31 mL/min/1.73 m^2^ (35.17), with five patients having augmented renal clearance (ARC). Most patients had only one sample analysed during their treatment, and the overall survival rate during the ICU admission was 71% (92 patients).

Detailed baseline demographic and clinical characteristics are presented in Table 1. ICU length of stay and the duration of mechanical ventilation (MV) were significantly higher in the ECMO group than in the control group. All admissions in the ECMO cohort were for medical conditions, mostly occurring during the SARS-CoV-2 pandemic, with no neurocritical cases. Most patients (83.3%) received veno-venous ECMO support, namely, ten and five patients in the piperacillin and meropenem cohorts, respectively (Appendix A).

A total of 222 antibiotic samples were collected under steady-state conditions, including 79 meropenem and 143 piperacillin plasma samples. Differences in clinical, analytical, infectious, and PK/PD parameters between the groups by sample are summarised in Appendix A. Boxplots of the *f*C_ss_ and *f*C_ss_/MIC ratio are shown in Figure 1. Only 48.50% of patients had a positive culture with a pathogen isolation (63/130), and the actual MIC was only available in about 50% of these samples (Appendix A).

The bottom and top extremes of the box represent the first (Q1) and the third quartile (Q3) range of the data, respectively (Q3–Q1: interquartile range). The dark horizontal line in the box is the median. ECMO = extracorporeal membrane oxygenation. MIC = minimum inhibitory concentration. *f*C_ss_ = free plasma concentration at steady state. *f*C_ss_/MIC = ratio of *f*C_ss_ to MIC.

Patients in the ECMO group had lower albumin and creatinine levels than controls, higher rates of directed antibiotic therapy for respiratory tract infections or bacteraemia, and a greater likelihood of MV. Despite these differences, patients on ECMO had significantly lower drug exposure, as measured by *f*C_ss_ and area under the curve (*f*AUC), in both the meropenem and piperacillin subgroups. However, no significant differences were observed in the *f*C_ss_/MIC ratio between groups. When assessing the proportion of episodes achieving PK/PD targets, approximately 70% of the meropenem episodes in both groups reached the PK/PD target of 100%*f*T_>4xMIC_ (78.58% in the control group and 65.38% in the ECMO group), but only 20% of the piperacillin episodes achieved the PK/PD target (20% in the control group and 18.60% in the ECMO group).

Univariate analysis revealed that MV, eGFR > 90 mL/min/1.73 m^2^, and ECMO were associated with significantly higher meropenem and piperacillin clearance and lower exposure (as assessed by *f*C_ss_, *f*C_ss_/daily dose, and *f*AUC). Notably, patients with ARC exhibited nearly half the *f*C_ss_ and *f*C_ss_/daily dose of those without ARC, across both meropenem and piperacillin cohorts. Multivariate analysis retained only eGFR and ECMO support in the meropenem model and eGFR and MV in the piperacillin model as independent clinical factors influencing antibiotic exposure. Detailed results of univariate and multivariate analyses are presented in Table 2 and Table 3.

## 3. Discussion

This MEPIMEX study provides novel insights into the PK behaviour of β-lactams in critically ill patients receiving ECMO support. It is the first to compare the PK/PD of β-lactams administered via continuous infusion in patients undergoing ECMO with eGFR > 60 mL/min/1.73 m^2^. Our results revealed that in patients treated with meropenem, PK/PD differences were predominantly influenced by renal function and ECMO support, while in those receiving piperacillin, renal function and MV played the most significant role. These findings suggest that renal function and MV appear to have a greater impact on drug exposure than ECMO. However, larger prospective studies are required to confirm this observation.

Preclinical studies examining ECMO-related alterations have consistently suggested that drugs with high protein binding and lipophilicity are more prone to sequestration [34]. In our study, meropenem, characterised by minimal protein binding (2%) and low lipophilicity [31], exhibited ECMO-related PK changes, diverging from most previously published studies [32,33,35,36,37,38]. Of note, ECMO has also been shown to accelerate meropenem degradation within the circuit, possibly attributable to the drug’s instability during blood exteriorisation [39,40]. A plausible explanation for the discrepancies observed in our study may relate to the continuous infusion strategy employed, as opposed to the intermittent infusion regimen reported in prior studies, which could have mitigated the spontaneous degradation of the antibiotic within the ECMO circuit. Beyond drug sequestration, additional mechanisms including altered capillary penetration, fluid shifts, and fluid retention present in ECMO patients may alter the volume of distribution and, consequently, could influence drug exposure, as previously described [41]. Notably, renal function remained the principal determinant of meropenem exposure, consistent with previous studies that have highlighted its importance [38,39,40]. Additionally, the low probability of target attainment observed in our cohort is likely attributable to the stringent PK/PD target, consistent with prior published studies [38,39,42].

Conversely, piperacillin, with a low level of protein binding (20%) and a hydrophilic profile, may have less predictable behaviour in the ECMO setting compared to meropenem [35,43]. Fewer than 50% of patients in both control and ECMO groups achieved the PK target. This was consistent with prior findings; however, in our study, the proportion of patients failing to achieve the PK/PD target was higher than previously reported, at approximately 60–40% [20,24,33,35]. The low attainment rates in our groups likely reflect the lack of microbiological isolation. The use of the theoretical MIC for piperacillin to assess PK/PD target attainment may overestimate the actual MIC of *Pseudomonas aeruginosa* in clinical isolates and overstate the rate of target failure.

Our study showed that piperacillin *f*C_ss_ was primarily influenced by renal function and the presence of MV, regardless of ECMO support. Most studies concur that ECMO does not significantly affect piperacillin *f*C_ss_, with target attainment largely dependent on creatinine clearance [29,30,33,35,44]. MV has been reported to influence cardiac output as well as renal, hepatic, and splanchnic perfusion, in addition to modifying the intrathoracic pressure, thereby leading to changes in PK parameters [45]. Although previous studies have suggested that MV may increase the volume of distribution and reduce the clearance of hydrophilic drugs, no statistically significant effects have been demonstrated for β−lactams [11,21]. These findings reinforce the need for TDM to account for inter-individual variability and to optimise dosing in critically ill patients.

Our study has several strengths. It is one of the largest and most homogeneous cohorts described to date focusing on ECMO patients with preserved renal function, treated under a standardised continuous infusion protocol. This approach minimises the heterogeneity seen in previous studies, which included mixed ECMO modes (veno-venous and veno-arterial) [30,31,35,38,44,46,47] and patients requiring renal replacement therapy [20,38,42] or intermittent antibiotic regimens [29,38,42,46].

Nevertheless, the study has some limitations that should be acknowledged. The main limitations are the small sample size, particularly in the ECMO cohort, in which most patients were included during the SARS-CoV-2 pandemic, the imbalance between groups, and the single-centre design. These can limit the generalisability of our findings and may have reduced the statistical power to detect clinically relevant differences. The temporal mismatch is also an important limitation because it introduces a potential bias due to differences in case mix, ICU practices, and pathogen distribution between periods. Other confounding factors related to severity of illness and type of patient not captured by the variables analysed may also have influenced the results obtained. Additionally, the retrospective nature of the study and the use of unmatched cohorts may have introduced potential bias. Moreover, the use of a surrogate MIC for piperacillin because only half of the patients had a positive culture with a pathogen isolation may have overestimated the PK/PD target failure rate, especially for microorganisms with lower MICs. Future studies incorporating organism-specific MICs in whole samples may provide a more accurate assessment of PK/PD attainment. Free concentrations were calculated using a fixed unbound fraction, which could underestimate patients’ real C_ss_, especially in those with hypoalbuminemia. Finally, a population PK modelling approach combined with Montecarlo simulations represents the gold standard for assessing the impact of clinical covariates on drug PK and accurately describes the elimination, but it requires rich data.

In our study, patients undergoing ECMO with both meropenem and piperacillin displayed significantly higher elimination with no significant differences in PK/PD target attainment between ECMO and non-ECMO groups. These alterations may reflect the severity of illness in ECMO-treated patients rather than the ECMO therapy itself [29,33]. For both drugs, continuous infusion provided *f*C_ss_ that remained above the MIC breakpoint for the isolated microorganism 100% of the time. For meropenem, our findings suggest that the standard dose of 3 g per day may be adequate for patients with ARC on ECMO, while lower doses may be suitable for those without ARC or ECMO support [20,39,48]. For piperacillin, the results highlight the need for caution when treating less susceptible microorganisms, as more aggressive targets (e.g., *f*C_ss_ > 32 or 64 mg/L) were not consistently achieved. The high inter-individual variability underscores the need for individualised therapeutic strategies and supports role of routine TDM to achieve optimal PK/PD targets in high-risk populations.

## 4. Materials and Methods

### 4.1. Study Design and Setting

The MEPIMEX study was a retrospective, single-centre, observational study conducted in the 90-bed adult ICU at Bellvitge University Hospital in Barcelona, Spain. The study included all adult ICU patients treated with a continuous infusion of meropenem or piperacillin while undergoing ECMO therapy between 1 April 2020, and 1 July 2021 (the ECMO group). This cohort was compared with a historical control group of critically ill patients who received continuously administered piperacillin or meropenem without ECMO support, derived from a previous study conducted in the same ICU between June 2015 and September 2018 [43]. Data collection and analytical procedures were identical in both groups to ensure comparability, and no changes were made in TDM hospital protocols between periods.

### 4.2. Eligibility Criteria

Inclusion criteria in the ECMO group were as follows: (i) age ≥ 18 years; (ii) diagnosis of sepsis or septic shock according to the Surviving Sepsis Campaign Guidelines [49]; (iii) treatment with meropenem or piperacillin administered as continuous infusion; and (iv) receiving ECMO therapy, regardless of the indication or technique used. Control group inclusion criteria have been described elsewhere [43].

Exclusion criteria for both groups were as follows: (i) pregnancy and (ii) impaired renal function, defined as eGFR < 60 mL/min/1.73 m^2^ using the 2009 Chronic Kidney Disease Epidemiology Collaboration (CKD-EPI) creatinine formula or requiring renal replacement therapy. Neither microbiological findings nor the initial antibiotic dose influenced patient inclusion.

### 4.3. Data Collection and Definitions

All the data were collected from electronic medical charts. Variables recorded included demographic, clinical, biochemical, haematological, and therapeutic and ECMO-related parameters.

The patient samples included in the ECMO group were defined as those concomitantly receiving ECMO and the antimicrobial for at least 24 h.

Neurocritical care patients were considered those with traumatic brain injury, acute ischemic stroke, intracerebral haemorrhage, or subarachnoid haemorrhage, conditions associated with a higher risk of developing ARC.

The eGFR was calculated from the serum creatinine concentrations according to the 2009 CKD-EPI creatinine formula. ARC was defined as an eGFR higher than 120 mL/min/1.73 m^2^ in women and 130 mL/min/1.73 m^2^ in men. Serum creatinine was measured using molecular absorption spectrometry.

### 4.4. β-Lactam Treatment and Measurements

The antimicrobial treatment was selected by the critical care physician in charge of the patient. The clinical indication of antimicrobial therapy could be either empirical or directed. Patients received an initial loading dose of 4 g piperacillin and 0.5 g tazobactam, infused over 30 min and immediately followed by a continuous infusion of 12 g piperacillin and 1.5 g tazobactam in 150 mL 0.9% sodium chloride (80 mg/mL, stable for 24 h at 25 °C). Those under meropenem received a loading dose of 1 g administered for 30 min, followed by a continuous infusion of 3 g of meropenem administered in two infusions per day (22 mg/mL in 0.9% saline, stable for 17 h at 25 °C).

Dose adjustments were made afterwards to achieve the targeted PK/PD of 100%*f*T_>4xMIC_ as per hospital protocol and following the latest international guidelines [50]. Toxicity thresholds were defined as *f*C_ss_ greater than 44 mg/L for meropenem and *f*C_ss_ greater than 156 mg/L for piperacillin. From a clinical perspective, pathogens with meropenem MIC > 8 mg/L and piperacillin MIC > 16 mg/L are rarely treated with these antibiotics.

To ensure steady-state conditions, samples were collected at least 24 h after the beginning of the antibiotic therapy or after any dose modification. Samples were centrifuged at 3000 rpm at 4 °C for 10 min, and the supernatant was removed and analysed by the ultra-performance liquid chromatography–tandem mass spectrometry technique (UPLC–MS/MS) [51]. Historical control samples were stored at -75 ± 3 °C for a maximum of 6 months before the analysis.

The tazobactam concentrations were not assessed. The total amount of antibiotic was quantified and free antibiotic concentration was calculated using the published unbound fraction (fu) of 0.3 for piperacillin and 0.02 for meropenem.

### 4.5. PK and PD Parameters

The following PK/PD parameters were calculated:

1.Unbound plasma clearance (CLu) [L/h] = daily dose [mg]/24 h · *f*C_ss_^−1^ [mg/L].2.*f*AUC_ss_ [mg·h/L] = daily dose [mg]/CLu [L/h].3.Dose-normalised C_ss_ = *f*C_ss_ [mg/L]/daily dose [mg].

We assumed a stable *f*C_ss_ through the whole dosing period (100% of the time) because antibiotics were administered in continuous infusion. We calculated the ratio *f*C_ss_/MIC (=C_ss_ [mg/dL] · fu/MIC) to determine whether *f*C_ss_ values were above the MIC (100%*f*T_>MIC_) and the number of times the MIC was achieved.

For patients with an isolated pathogen, actual MIC values were used for PK/PD calculations. Otherwise, inferred theoretical MIC breakpoints were used: 2 mg/L for meropenem and 16 mg/L for piperacillin, as recommended by the European Committee on Antimicrobial Susceptibility Testing (EUCAST) [52].

Antibiotic exposure was assessed by *f*C_ss_ and *f*AUC_ss_ achieved. This allowed us to determine differences in exposure achieved without bias due to differences in MIC distributions among ICUs and actual and surrogate MIC values.

### 4.6. Statistical Analysis

Continuous variables are summarised as means and standard deviations (SD), while categorical variables are presented as frequencies and percentages. Univariate analyses were performed to identify covariates significantly associated with meropenem and piperacillin *f*C_ss_. The association between categorical and continuous variables was determined using Student’s *t*-test and one-way analysis of variance (F distribution). The association between continuous variables was stated by using simple linear regression models. Significant variables were subsequently included in multivariate analyses. Generalised estimating equations (GEEs) were applied to study the association between *f*C_ss_ and previously statistically significant variables obtained in the univariate analysis, as they allowed analysis of repeated measures. Factors included were assumed to be categorical variables, and cofactors were assumed to be continuous. Logarithmic transformation of the dependent variable was performed. Mixed-effects models were not used due to the absence of scale variables. Statistical significance was set at *p* < 0.05. Statistical analyses were conducted using SPSS Statistics (version 30, SPSS Inc., an IBM Company, Chicago, IL, USA).

## 5. Conclusions

Our study reveals significant variability in β-lactam exposure and PK/PD target attainment, influenced primarily by renal function and ECMO in patients under meropenem and by renal function and MV in patients receiving piperacillin. These findings emphasise the critical importance of TDM and personalised dosage strategies to enhance outcomes in critically ill patients. Our results should be interpreted with caution due to the limited sample size, and further prospective research is essential to validate these results in larger, more diverse populations and across varied geographical settings.

## Figures and Tables

**Figure 1 antibiotics-14-00939-f001:**
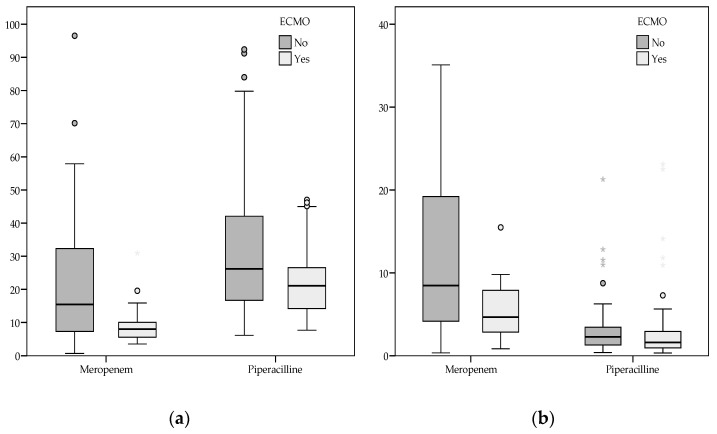
Boxplot of meropenem and piperacillin (**a**) *f*C_ss_ and (**b**) *f*C_ss_/MIC ratio sorted by group.

**Table 1 antibiotics-14-00939-t001:** Baseline demographics, clinical and microbiological characteristics of the patients included in the study.

	Control Group	ECMO Group	*p*-Value
Meropenem			
Number of patients	40 (85.1)	7 (14.9)	
Sex (male/female)	26 (65)/14 (35)	7 (100)/0 (0)	0.086
Age, year	59.4 (13.5)	60.6 (18.4)	0.417
Weight, kg	74.34 (16.9)	85.1 (9.9)	0.540
BMI, kg/m^2^ Underweight (≤18.5) Normal weight (18.6–24.9) Overweight (25–29.9) Obese (≥30)	3 (7.50) 12 (30.00) 14 (35.00) 11 (27.50)	0 (0) 0 (0) 5 (71.43) 2 (28.57)	0.205
Baseline eGFR	85.1 (21.9)	89.5 (5.5)	0.162
SOFA scale value	6.40 (3.28)	6.57 (1.90)	0.447
Neurocritical patient	6 (15.0)	0 (0)	0.571
Admission diagnosis Surgical Medical Trauma	16 (40.0) 19 (47.5) 5 (12.50)	0 (0) 7 (100) 0 (0)	**0.036**
Days of hospitalisation	51.9 (44.6)	45.3 (24.4)	0.351
Days in ICU	20.5 (20.9)	37.7 (20.9)	**0.025**
Days of MV	11.1 (13.5)	39.9 (18.4)	**<0.001**
Outcome (exitus)	10 (25.00)	4 (57.14)	0.173
**Piperacillin**			
Number of patients	72 (86.75)	11 (13.25)	
Sex (male/female)	45 (62.50)/27 (37.50)	9 (81.82)/2 (18.18)	0.211
Age, year	60.4 (16.1)	72.0 (12.5)	**0.012**
Weight, kg	75.9 (17.9)	88.6 (24.2)	**0.020**
BMI, kg/m^2^ Underweight (≤18.5) Normal weight (18.6–24.9) Overweight (25–29.9) Obese (≥30)	3 (4.17) 25 (34.72) 26 (36.11) 18 (25.00)	0 (0) 3 (27.27) 4 (36.36) 4 (36.36)	0.782
Baseline eGFR	91.5 (20.9)	83.5 (12.5)	0.112
SOFA scale value	5.63 (3.16)	6.00 (4.36)	0.364
Neurocritical patient	22 (30.6)	0 (0)	**0.032**
Admission diagnosis Surgical Medical Trauma	34 (47.22) 32 (44.44) 6 (8.33)	0 (0) 11 (100) 0 (0)	**0.003**
Days of hospitalisation	45.9 (41.4)	85.7 (43.3)	**0.002**
Days in ICU	19.9 (17.2)	70.6 (32.2)	**<0.001**
Days of MV	20.9 (86.2)	67.2 (27.4)	**0.041**
Outcome (exitus)	19 (26.39)	5 (45.46)	0.194

Expressed as means (standard deviation). Bold *p*-values represent statistical significance (*p* < 0.05). Abbreviations: BMI = body mass index. ECMO = extracorporeal membrane oxygenation. eGFR = estimated glomerular filtration rate (mL/min/1.73 m^2^). ICU = intensive care unit. MV = mechanical ventilation.

**Table 2 antibiotics-14-00939-t002:** Effect of covariates on meropenem and piperacillin exposure and pharmacokinetic parameters.

Covariate		*f*C_ss_ (mg/L)	*f*C_ss_ (mg/L)/ Daily Dose	lnCL_u_ (L/h)	*f*AUC (mg·h/L)	*p*-Value
Meropenem						
Sex	Male (59) Female (20)	17.4 (17.2) 18.9 (18.1)	5.8 (5.7) 6.3 (6.1)	2.38 (0.94) 2.26 (0.86)	419.2 (412.1) 453.8 (436.1)	0.615
Neurocritical status	Yes (11) No (68)	21.8 (14.5) 17.2 (17.8)	7.3 (4.8) 5.7 (5.9)	1.99 (0.77) 2.40 (0.93)	522.8 (347.2) 412.6 (426.0)	0.169
Post-surgical drainage	Yes (31) No (48)	16.2 (14.9) 18.9 (18.8)	5.4 (4.9) 6.3 (6.3)	2.49 (1.01) 2.26 (0.85)	387.9 (359.3) 453.8 (450.2)	0.285
MV	Yes (56) No (23)	15.2 (14.1) 24.2 (22.4)	5.1 (4.7) 8.1 (7.5)	2.50 (0.92) 1.97 (0.82)	365.3 (339.7) 580.5 (538.4)	**0.019**
Vasoactive drugs	Yes (26) No (53)	18.9 (20.8) 17.3 (15.5)	6.3 (6.9) 5.7 (5.2)	2.33 (0.94) 2.36 (0.91)	454.8 (498.6) 414.8 (373.0)	0.908
Admission diagnosis	Surgical (20) Medical (51) Trauma (8)	18.2 (17.0) 16.9 (18.1) 22.8 (12.9)	6.1 (5.7) 5.6 (6.1) 7.6 (4.3)	2.47 (1.18) 2.37 (0.83) 1.88 (0.68)	432.3 (408.9) 406.0 (435.6) 546.8 (311.3)	0.301
BMI	≤18.5 (3) 18.6–24.9 (12) 25–29.9 (18) ≥30 (13)	17.4 (7.2) 34.0 (28.1) 21.1 (17.1) 16.3 (14.8)	5.8 (2.4) 11.3 (9.4) 6.9 (5.8) 5.4 (4.9)	2.03 (0.44) 1.61 (0.84) 2.32 (1.27) 2.38 (0.86)	417.9 (171.7) 816.9 (673.4) 506.9 (410.4) 391.2 (356.3)	0.230
eGFR	60–89 (16) 90–119/129 (40) ≥120/130 (23)	33.4 (25.5) 15.7 (13.0) 10.7 (9.1)	11.1 (8.5) 5.2 (4.4) 3.6 (3.0)	1.64 (0.90) 2.44 (0.93) 2.68 (0.63)	800.7 (611.2) 377.4 (313.1) 256.6 (218.5)	**0.001**
ARC	60–119/129 (56) ≥120/130 (23)	20.8 (19.0) 10.7 (9.1)	6.9 (6.4) 3.6 (3.0)	2.21 (0.98) 2.68 (0.63)	498.3 (456.8) 256.6 (218.5)	**0.014**
ECMO	Yes (26) No (53)	9.4 (5.8) 21.9 (19.5)	3.1 (1.9) 7.3 (6.5)	2.76 (0.61) 2.14 (0.98)	226.6 (138.8) 526.7 (468.8)	**0.001**
**Piperacillin**						
Sex	Male (102) Female (41)	28.3 (18.7) 32.6 (23.9)	2.3 (1.6) 2.7 (1.9)	3.10 (0.62) 2.94 (0.64)	680.5 (448.3) 783.5 (572.8)	0.165
Neurocritical status	Yes (31) No (112)	25.0 (14.2) 30.8 (21.6)	2.1 (1.2) 2.5 (1.8)	3.15 (0.59) 3.03 (0.63)	600.9 (341.3) 740.2 (517.9)	0.315
Post-surgical drainage	Yes (37) No (106)	29.5 (19.9) 29.6 (20.5)	2.4 (1.7) 2.4 (1.7)	3.07 (0.67) 3.05 (0.61)	709.1 (478.8) 710.3 (492.7)	0.820
MV	Yes (96) No (47)	24.8 (14.6) 39.4 (26.2)	1.9 (1.2) 3.3 (2.2)	3.21 (0.57) 2.73 (0.62)	594.7 (349.4) 945.6 (629.9)	**<0.001**
Vasoactive drugs	Yes (39) No (104)	28.2 (17.2) 30.1 (21.4)	2.2 (1.4) 2.5 (1.8)	3.12 (0.60) 3.02 (0.63)	676.0 (413.5) 722.8 (513.7)	0.438
Admission diagnosis	Surgical (48) Medical (88) Trauma (7)	33.6 (19.9) 28.6 (20.7) 15.0 (4.5)	2.8 (1.7) 2.3 (1.7) 1.3 (0.4)	2.88 (0.63) 3.11 (0.61) 3.56 (0.39)	805.7 (479.4) 658.6 (497.4) 360.7 (107.1)	**0.01**
BMI	≤18.5 (3) 18.6–24.9 (28) 25–29.9 (31) ≥30 (22)	52.1 (24.1) 35.1 (25.5) 30.2 (23.8) 31.6 (16.9)	4.3 (2.0) 2.9 (2.1) 2.5 (1.9) 2.6 (1.4)	2.32 (0.43) 2.92 (0.75) 3.01 (0.61) 2.90 (0.54)	1250.5 (577.5) 842.8 (612.1) 721.8 (571.4) 757.9 (406.1)	0.372
eGFR	60–89 (38) 90–119/129 (74) ≥120/130 (31)	41.8 (24.8) 28.2 (18.2) 17.9 (7.1)	3.5 (2.1) 2.3 (1.5) 1.4 (0.6)	2.62 (0.53) 3.09 (0.61) 3.48 (0.39)	1002.25 (594.5) 677.2 (438.3) 430.0 (171.6)	**<0.001**
ARC	60–119/129 (114) ≥120/130 (29)	32.6 (21.5) 17.9 (7.3)	2.7 (1.8) 1.4 (0.6)	2.95 (0.63) 3.48 (0.41)	781.4 (515.4) 429.2 (174.8)	**<0.001**
ECMO	Yes (43) No (100)	23.6 (14.8) 32.1 (21.8)	1.8 (1.2) 2.7 (1.8)	3.30 (0.49) 2.95 (0.65)	566.8 (356.2) 771.6 (523.7)	**0.001**

Expressed as means (standard deviation). Bold *p*-values represent statistical significance (*p* < 0.05). *p*-value was the same for *f*C_ss_/MIC, *f*AUC, and CLu. *f*C_ss_/MIC, CLu, and *f*AUC log-transformed values. ARC values were defined first for men and second for women. Abbreviations: ARC = augmented renal clearance. BMI = body mass index. ECMO = extracorporeal membrane oxygenation. CLu = unbound antibiotic clearance. eGFR = estimated glomerular filtration rate (mL/min/1.73 m^2^). *f*AUC = free area under the curve. *f*C_ss_ = free antibiotic concentrations. F = female. M = male. MV = mechanical ventilation.

**Table 3 antibiotics-14-00939-t003:** Multivariate analysis of covariates’ influences on meropenem and piperacillin exposure.

	Covariate	Beta	95% CI	*p*-Value
	eGFR ECMO MV	0.070	0.001–0.012	**0.014**
Meropenem		0.453	0.015–0.891	**0.043**
		0.089	−0.386–0.565	0.713
	eGFR ECMO MV	0.130	0.002–0.009	**<0.001**
Piperacillin		−0.187	−0.406–0.032	0.095
		0.313	0.0577–0.200	**<0.001**

Bold *p*-values represent statistical significance (*p* < 0.05). Abbreviations: CI = confidence interval. eGFR = estimated glomerular filtration rate (mL/min/1.73 m^2^). ECMO = extracorporeal membrane oxygenation. MV = mechanical ventilation.

## Data Availability

The datasets generated and/or analysed during the current study are available from the corresponding author on reasonable request.

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
