# Peer review of "Optimising Meropenem and Piperacillin Dosing in Patients Undergoing Extracorporeal Membrane Oxygenation Without Renal Dysfunction (MEPIMEX)"

_antibiotics, 2025, doi:10.3390/antibiotics14090939_

Round 1
Reviewer 1 Report
Comments and Suggestions for Authors
I have read this paper with a background on clinical research, including PK/PD of antimicrobials during ECMO. There is value in the paper, while additional information is warranted in my opinion
Abstract
Line 30: ‘may be’ is very likely too soft statement, as this is very commonly the case, related to either Vd or Cl alterations. Can you add a line on the PD targets in the methods section, if word count allows.
General
To the best of my understanding, there is a potential issue on meropenem stability published in this journal (PMID 33923550) after collection and storage. I suggest the authors to consider this literature (I’m not involved in the PMID) and to revise as and if appropriate (control group seems to be based on historical data, but have samples been stored previously). Additional questions on the control group are provided lower
Any advice on how to address this issue on dosing. Do we need an additional loading type of dose at initiation of ecmo for these water soluble tested ? I understand that you have collected at steady state, but assume that the time interval following ecmo initiation is relevant ?
Specific
How has eGFR been measured and what assay has been used to quantify creatinine (or cystatin).
We need more information on the control group.
Where samples simultaneously collected in ecmo cases and controls ? was this part of a TDM practice in your hospital ?
Why have you excluded impaired renal function, since this limits the clinical ‘relevance’ of your findings quite significant. E.g. how many cases were excluded because of impaired renal function ? This should at least be clearer reflected in the abstract/title.
The antibiotic assays needs reference(s)
Line 187 onwards: these aspects are not simply mutual exclusive. Besides adherence to the circuit, albumin, renal function and Vd changes are likely also contributing factor to the final phenotype. Suggest to rephrase.
Type of ecmo (vv, va, type of pump), and materials are useful information.
Author Response
Dear Editor,
We would like to express our sincere gratitude to you and the reviewers for the careful evaluation of our manuscript and for the valuable comments provided. We have revised the paper accordingly, and we believe these changes have significantly improved the clarity, accuracy, and overall quality of the work.
Below we provide a detailed, point-by-point response to each of the reviewers’ comments. For clarity, the reviewers’ comments are presented in black letter, followed by our responses in blue. All modifications have been incorporated into the revised version of the manuscript, you can find it in red in the marked version. The lines mentioned in the corrections refer to those of the revised version of the manuscript with the changes marked in red (MEPIMEX study_marked version).
Response to first reviewer comments
|
Yes |
Can be improved |
Must be improved |
Not applicable |
|
|
Does the introduction provide sufficient background and include all relevant references? |
(x) |
( ) |
( ) |
( ) |
|
Is the research design appropriate? |
(x) |
( ) |
( ) |
( ) |
|
Are the methods adequately described? |
( ) |
( ) |
(x) |
( ) |
|
Are the results clearly presented? |
( ) |
(x) |
( ) |
( ) |
|
Are the conclusions supported by the results? |
( ) |
(x) |
( ) |
( ) |
|
Are all figures and tables clear and well-presented? |
(x) |
( ) |
( ) |
( ) |
Thank you for the suggestions, we have tried to improve the methods’ section and describe them in a clearer way.
*Comments and Suggestions for Authors
I have read this paper with a background on clinical research, including PK/PD of antimicrobials during ECMO. There is value in the paper, while additional information is warranted in my opinion
- Abstract - Line 30: ‘may be’ is very likely too soft statement, as this is very commonly the case, related to either Vd or Cl alterations.
Thank you for your suggestion. We have replaced ‘may be’ with ‘are’ (line 31 of the new manuscript).
- Abstract - Can you add a line on the PD targets in the methods section, if word count allows.
As the reviewer recommended, we have included a line on the PK/PD targets of both meropenem and piperacillin in the method section:
‘For both antimicrobials, PK/PD target was set at 100%fT>4xMIC’ (line 39 and 40 of the new manuscript).
- General - To the best of my understanding, there is a potential issue on meropenem stability published in this journal (PMID 33923550) after collection and storage. I suggest the authors to consider this literature (I’m not involved in the PMID) and to revise as and if appropriate (control group seems to be based on historical data, but have samples been stored previously).
Thank you for providing such information. Drug concentrations were determined using the previously described ultra-performance liquid chromatography-tandem mass spectrometry technique (UPLC–MS/MS), which was developed in our hospital (Clin Chim Acta. 2017;468:215–24, https://doi.org/10.1016/j.cca.2017.03.009). Control group samples were stored at -80ºC until determination for a maximum of 6 months. Our study demonstrated that β-lactam concentrations in plasma were stable at −75 ± 3 °C for at least 6 months (absolute %D ≤ 9.9%) which is confirmed by PMID 33923550. Gijsen M et al showed that the measured concentration for plasma samples that were stored during the complete study period at −80 °C (i.e., time at −20 °C = 0 days) was within 94–105% of the spiked concentration, thus showing no measurable degradation under these conditions.
We have clarified the methods section:
‘Historical control samples were stored at - 75 ± 3 °C for a maximum of 6 months before the analysis’ (lines 369 to 370 of the new manuscript).
- General - Additional questions on the control group are provided lower. We need more information on the control group:
- Where samples simultaneously collected in ECMO cases and controls?
Samples were not simultaneously collected in cases and controls. Non-ECMO control group was a historical cohort and results of the study were published previously (Ref 43, Eur J Drug Metab Pharmacokinet. 2021 Jul;46(4):527-538. DOI: 10.1007/s13318-021-00694-0). We have added this as a limitation in the discussion section.
‘The temporal mismatch is also an important limitation because it introduces a potential bias due to differences in case mix, ICU practices, and pathogen distribution between periods‘ (lines 278 to 280 of the new manuscript).
- Was this part of a TDM practice in your hospital?
Yes, it was. We have specified this in the text:
‘as per TDM hospital protocol’ (lines 357 of the new manuscript).
- General - Any advice on how to address this issue on dosing. Do we need an additional loading type of dose at initiation of ECMO for these water soluble tested? I understand that you have collected at steady state, but assume that the time interval following ECMO initiation is relevant?
We do not think that a new loading dose should be necessary after the initiation of ECMO therapy. Our suggestion is to improve the antibiotic exposure by using therapeutic drug monitoring.
Per protocol, to ensure that steady state conditions were achieved, we obtained samples after 24 hours of concomitant ECMO therapy and stable antimicrobial dose. There is no evidence whether time interval following ECMO initiation is relevant. However, we followed this strategy to avoid bias and the influence of unknown and non-predictable ECMO factors on drug concentrations.
References:
- DOI: 10.3390/jcm13123554.
- DOI: 10.1111/jcpt.12636
- DOI: 10.1097/ACO.0000000000000810
- DOI: 10.1002/phar.1882
- Specific - How has eGFR been measured and what assay has been used to quantify creatinine (or cystatin).
Estimated glomerular filtration rate (eGFR) was calculated using 2009 Chronic Kidney Disease Epidemiology Collaboration (CKD-EPI) creatinine formula using the creatinine serum values. The technique used to quantify the serum creatinine concentrations was molecular absorption spectrometry.
We have clarified it in the methods section:
‘Exclusion criteria for both groups were: (i) pregnancy and (ii) impaired renal function, defined as eGFR using 2009 Chronic Kidney Disease Epidemiology Collaboration (CKD-EPI) creatinine formula <60 mL/min/1.73m² or requiring renal replacement therapy. Neither microbiological findings nor the initial antibiotic dose influenced patient inclusion.’ (lines 326 to 330 of the new manuscript).
‘The eGFR was calculated from the serum creatinine concentrations according to 2009 CKD-EPI creatinine formula. ARC was defined as an eGFR higher than 120 mL/min/1.73m2 in women and 130 mL/min/1.73m2 in men. Serum creatinine was measured using molecular absorption spectrometry.’ (lines 341 to 344 of the new manuscript).
- Specific - Why have you excluded impaired renal function, since this limits the clinical ‘relevance’ of your findings quite significant. E.g. how many cases were excluded because of impaired renal function? This should at least be clearer reflected in the abstract/title.
Meropenem and piperacillin’s exposure are clearly influenced by the renal function. We designed our study to assess whether patients without renal dysfunction were at risk of antibiotic underexposure.
After patient inclusion and during the study period, 10 patient samples were finally excluded due to renal deterioration with eGFR <60 ml/min/1.73m2, 3 in the ECMO group and 7 in the non-ECMO group.
Accordingly, we have modified the title:
‘Optimising meropenem and piperacillin dosing in patients undergoing extracorporeal membrane oxygenation without renal dysfunction (MEPIMEX)’
We also have included the information in the abstract: ‘without renal dysfunction’ (line 37 of the new manuscript)
- Specific - The antibiotic assays needs reference(s)
We have included the reference for the antibiotic’s assays: reference number 51 - Rigo Bonnin R, Alía Ramos P. Desarrollo y validación de un procedimiento de medida para la medicación simultánea de la concentración de masa de ceftazidima, meropenem y piperacilina en el plasma mediante UHPLC-MS/MS. Rev Lab Clin. 2017, 10, 3-13. DOI: 10.1016/j.labcli.2016.10.004 (line 369 of the new manuscript).
- Specific - Line 187 onwards: these aspects are not simply mutual exclusive. Besides adherence to the circuit, albumin, renal function and Vd changes are likely also contributing factor to the final phenotype. Suggest to rephrase.
Thank you for your suggestion. We have rephrased the paragraph and added that besides drug sequestration, other variables may change volume of distribution and influence drug exposure.
‘Besides drug sequestration, capillary penetration, fluid shifts and retention present in ECMO patients are mechanisms that may alter volume of distribution and could also influence drug exposure as previously described [41].’ (lines 236 to 238 of the new manuscript).
- Specific - Type of ecmo (vv, va, type of pump), and materials are useful information.
Following the reviewer’s suggestions, we added more information of ECMO mode, membrane and blood pump. We have included this in the results section and as supplementary material.
‘Most patients (83.3%) received veno-venous ECMO support, 10 and 5 patients in the piperacillin and meropenem cohort respectively (Table S1).’ (lines 145 to 147 of the new manuscript).
‘Table S1. ECMO characteristics of the recruited patients’
|
Patient ID |
Drug |
Indication |
Duration |
ECMO mode |
Oxygenator membrane |
Bloodpump |
|
1 |
Piperacillin |
ARDS |
75 |
VV |
Maquet Rotaflow |
HLS |
|
8 |
Piperacillin |
ARDS |
123 |
VV |
Maquet Rotaflow |
PLS |
|
13 |
Piperacillin |
ARDS |
55 |
VV |
CardioHelp |
HLS |
|
18 |
Piperacillin |
ARDS |
25 |
VV |
Maquet Rotaflow |
PLS |
|
24 |
Meropenem |
ARDS |
21 |
VV |
CardioHelp |
HLS |
|
27 |
Piperacillin |
ARDS |
45 |
VV |
CardioHelp |
HLS |
|
35 |
Meropenem |
ARDS |
63 |
VV |
Centrimag |
HLS |
|
43 |
Piperacillin |
ARDS |
49 |
VV |
Maquet Rotaflow |
PLS |
|
48 |
Meropenem |
ARDS |
8 |
VV |
Maquet Rotaflow |
PLS |
|
52 |
Meropenem |
ARDS |
40 |
VV |
CardioHelp |
HLS |
|
55 |
Piperacillin |
ARDS |
40 |
VV |
CardioHelp |
HLS |
|
60 |
Meropenem |
ARDS |
43 |
VV |
Maquet Rotaflow |
PLS |
|
64 |
Piperacillin |
ARDS |
56 |
VV |
Maquet Rotaflow |
PLS |
|
67 |
Piperacillin |
ARDS |
29 |
VV |
CardioHelp |
HLS |
|
70 |
Piperacillin |
ARDS |
21 |
VV |
CardioHelp |
HLS |
|
72 |
Piperacillin |
ARDS |
23 |
VV |
CardioHelp |
HLS |
|
111 |
Meropenem |
Acute myocardial infarction |
11 |
VA |
CardioHelp |
HLS |
|
186 |
Meropenem |
Acute myocardial infarction |
13 |
VA |
CardioHelp |
HLS |
ARDS: Acute respiratory distress syndrome; VV: Veno-venous; VA: Veno-arterial.

Reviewer 2 Report
Comments and Suggestions for Authors
The manuscript addresses an important and timely clinical question: how extracorporeal membrane oxygenation (ECMO) affects β-lactam pharmacokinetics (PK) and pharmacodynamics (PD), specifically for meropenem and piperacillin. The study design—a retrospective observational comparison with a control cohort—is appropriate within its constraints, and the paper is generally well-structured, methodologically sound, and clinically relevant.
It makes a meaningful contribution by showing that renal function and mechanical ventilation may play a more significant role in antibiotic exposure than ECMO itself, thereby reinforcing the importance of individualized therapeutic drug monitoring (TDM).
That said, several areas need improvement. The presentation could be clearer, the conclusions should be expressed with greater caution, limitations should be emphasized more directly, and the English style could be refined for conciseness and readability.
Introduction
The introduction is well-written and provides solid background on β-lactam PK/PD and the challenges introduced by ECMO. However, there is some redundancy, particularly around the discussion of TDM, which is mentioned multiple times. Streamlining would improve readability without sacrificing content.
Study Design and Cohort
The ECMO sample size is very small (n=18), which significantly limits statistical power and generalizability. This limitation should be acknowledged more forcefully, and conclusions should be tempered accordingly.
There is also a temporal mismatch between cohorts: ECMO patients were recruited during the COVID-19 pandemic, while the control cohort predates it. This introduces potential bias due to differences in case mix, ICU practices, and pathogen distribution. These factors warrant more explicit discussion.
PK/PD Targets
The requirement of achieving 100% fT > 4–8× MIC is more aggressive than what is typically used and may exaggerate target failure rates. The authors should provide stronger clinical justification for this choice and discuss how it aligns with EUCAST and IDSA guidance.
Interpretation of Findings
Statements suggesting that ECMO “may not require dose adjustment” should be expressed more cautiously, given the small sample size and retrospective nature of the study.
The finding that mechanical ventilation was an independent determinant of clearance is intriguing, but it could also reflect illness severity as a confounding factor. This point deserves a more nuanced discussion.
Methodological Considerations
The use of a fixed unbound fraction (fu) for TDM calculations may underestimate variability in critically ill patients, especially those with hypoalbuminemia. This limitation should be emphasized more clearly.
Additionally, about half of the isolates lacked actual MIC data. Relying on surrogate MICs—particularly for piperacillin—may overstate the rate of PK/PD failure. This issue should be addressed in the discussion.
It would also be useful to specify whether all ECMO patients were on veno-venous or veno-arterial support, since ECMO mode can influence PK. Furthermore, the decision not to use mixed-effects modeling should be justified more clearly, as this may be questioned by readers familiar with pharmacometric methods.
Results
The results are comprehensive and well-documented, but the tables are dense and somewhat difficult to interpret. Some could be simplified or moved to supplementary material. Presenting selected results graphically—for example, using boxplots or forest plots of fCss/MIC ratios—would make the data more accessible and highlight key differences more effectively.
Discussion
The discussion integrates the findings with existing literature very well. However, it needs stronger critical reflection on possible confounding factors, such as the influence of the pandemic, differences in case severity, and selection bias.
The conclusion that ECMO may not require specific dose adjustments should be softened. A safer interpretation would be that renal function and mechanical ventilation appear to play a greater role than ECMO in shaping drug exposure, but that larger prospective studies are needed for confirmation.
Conclusion
The conclusion is concise and aligned with the study’s findings but should include a clearer cautionary statement about limited generalizability.
Comments on the Quality of English Language
Language & Style
Overall, the manuscript is readable, but the language would benefit from editing for clarity and conciseness. Several sentences are long and complex, which makes them harder to follow. Shortening these and eliminating redundancies would improve flow.
There are also occasional grammar issues (for example, “the throughout the dosing interval” should be “throughout the dosing interval”). Finally, word choices could be refined to maintain academic tone—for instance, “suffice” could be replaced with “may be adequate.”
Author Response
Dear Editor,
We would like to express our sincere gratitude to you and the reviewers for the careful evaluation of our manuscript and for the valuable comments provided. We have revised the paper accordingly, and we believe these changes have significantly improved the clarity, accuracy, and overall quality of the work.
Below we provide a detailed, point-by-point response to each of the reviewers’ comments. For clarity, the reviewers’ comments are presented in black letter, followed by our responses in blue. All modifications have been incorporated into the revised version of the manuscript, you can find it in red in the marked version. The lines mentioned in the corrections refer to those of the revised version of the manuscript with the changes marked in red (MEPIMEX study_marked version).
Response to second reviewer comments
Dear reviewer, we are very pleased to hear about your feedback on our manuscript. We will modify the text accordingly to each comment as best as we can.
|
Yes |
Can be improved |
Must be improved |
Not applicable |
|
|
Does the introduction provide sufficient background and include all relevant references? |
( ) |
(x) |
( ) |
( ) |
|
Is the research design appropriate? |
(x) |
( ) |
( ) |
( ) |
|
Are the methods adequately described? |
(x) |
( ) |
( ) |
( ) |
|
Are the results clearly presented? |
( ) |
(x) |
( ) |
( ) |
|
Are the conclusions supported by the results? |
( ) |
(x) |
( ) |
( ) |
|
Are all figures and tables clear and well-presented? |
( ) |
(x) |
( ) |
( ) |
Comments and Suggestions for Authors
The manuscript addresses an important and timely clinical question: how extracorporeal membrane oxygenation (ECMO) affects β-lactam pharmacokinetics (PK) and pharmacodynamics (PD), specifically for meropenem and piperacillin. The study design—a retrospective observational comparison with a control cohort—is appropriate within its constraints, and the paper is generally well-structured, methodologically sound, and clinically relevant.
It makes a meaningful contribution by showing that renal function and mechanical ventilation may play a more significant role in antibiotic exposure than ECMO itself, thereby reinforcing the importance of individualized therapeutic drug monitoring (TDM).
That said, several areas need improvement. The presentation could be clearer, the conclusions should be expressed with greater caution, limitations should be emphasized more directly, and the English style could be refined for conciseness and readability.
- Introduction - The introduction is well-written and provides solid background on β-lactam PK/PD and the challenges introduced by ECMO. However, there is some redundancy, particularly around the discussion of TDM, which is mentioned multiple times. Streamlining would improve readability without sacrificing content.
Thank you very much for the comment. We have re-read the introduction and there is certainly some redundancy in the text. Specifically involving the purposed strategies to overcome resistance to beta-lactam antibiotics and achieve the PK/PD target in intensive care patients. We have modified the whole introduction section.
- Study Design and Cohort - The ECMO sample size is very small (n=18), which significantly limits statistical power and generalizability. This limitation should be acknowledged more forcefully, and conclusions should be tempered accordingly.
Thank you for the observation. In the multivariable models used, in order to minimise the potential loss of power and to ensure the normality of the studied samples, the natural logarithmic transformation of the independent variable was applied (Y = Ln(fc)).
On the other hand, normality tests (Kolmogorov–Smirnov) for the study subsamples of piperacillin and meropenem showed that normality was not violated for the numerical variables included in the model.
In this context, the inclusion of the ECMO variable in both models — statistically significant in the meropenem substudy and with a trend towards significance in the piperacillin substudy — highlights the impact of ECMO on the dependent variable.
Secondly, and as suggested, we have mentioned this limitation more forcefully in the discussion section and have tempered the conclusions.
Discussion section:
‘Nevertheless, the study has some limitations that should be acknowledged. The main limitations are the small sample size, particularly in the ECMO cohort in which most patients were included during the SARS-Cov-2 pandemic, the imbalance between groups and the single-center design. These can limit the generalisability of our findings and may have reduced the statistical power to detect clinically relevant differences.’ (lines 272 to 277 of the new manuscript).
Conclusion section:
‘Our results should be interpreted with caution due to the limited sample size and further prospective research is essential to validate these results in larger, more diverse populations and across varied geographical settings.’ (lines 418 to 420 of the new manuscript).
- Study Design and Cohort - There is also a temporal mismatch between cohorts: ECMO patients were recruited during the COVID-19 pandemic, while the control cohort predates it. This introduces potential bias due to differences in case mix, ICU practices, and pathogen distribution. These factors warrant more explicit discussion.
We agree with the reviewer, and we have included this as a limitation in the discussion section.
‘The temporal mismatch is also an important limitation because it introduces a potential bias due to differences in case mix, ICU practices, and pathogen distribution between periods’ (lines 280 to 282 from the new manuscript).
However, we had considered this limitation in the statistical analysis and we want to point out that:
- Baseline demographic variables did not show significant differences between the two subgroups.
- Between both periods of time, no changes were made in the protocols of how to use the piperacillin or the meropenem (frequency of administration, dose, time of perfusion, dilution neither time to plasma concentration recollection) and TDM and dose adjustment.
In order to improve clarity, we have included a sentence explicitly addressing this point:
‘Data collection and analytical procedures were identical in both groups to ensure comparability, and no changes were made in TDM hospital protocols between periods.’ (lines 318 to 320 from the new manuscript).
- PK/PD Targets - The requirement of achieving 100% fT > 4–8× MICis more aggressive than what is typically used and may exaggerate target failure rates. The authors should provide stronger clinical justification for this choice and discuss how it aligns with EUCAST and IDSA guidance.
The PK/PD target of 100%fT>4 x MIC is the most aggressive and optimum therapeutic objective based on clinical assays and meta-analysis in the ICU setting and current guidelines. This target has been related with better clinical cure rates and lower risk of resistance development, especially in severe gram-negative bacterial infections.
Included in the IDSA practice guidelines (https://www.idsociety.org/practice-guideline/accp_prolonged_infusion_beta_lactums/), an international consensus recommendations for the use of prolonged-infusion beta-lactam antibiotics developed by the American College of Clinical Pharmacy, British Society for Antimicrobial Chemotherapy, Cystic Fibrosis Foundation, European Society of Clinical Microbiology and Infectious Diseases, Infectious Diseases Society of America, Society of Critical Care Medicine, and Society of Infectious Diseases Pharmacists was published a in 2023. The panel concluded that the current evidence is insufficient to recommend a specific concentration or universal exposure target when performing β-lactam TDM. Nevertheless, for β-lactams administered as continuous infusion, the panel suggests targeting 100% fT>MIC with concentrations maintained at least 4 times above the MIC (DOI: https://doi.org/10.1002/phar.2842). We have added the new reference in the manuscript, reference [50] (line 359 from the new manuscript)
Additional references that also support the aggressive PK/PD target in our daily clinical practice:
- DOI: 10.1186/s13054-020-03272-z
- DOI: 10.1186/s13054-024-04911-5
- DOI: 10.1080/14787210.2017.1338139
- DOI: 10.1093/jac/dkaf013
- Interpretation of Findings - Statements suggesting that ECMO “may not require dose adjustment” should be expressed more cautiously, given the small sample size and retrospective nature of the study.
Thank you very much, we have replaced the sentence in the first paragraph of the discussion section with a softened interpretation.
‘These findings suggest that renal function and MV appear to have a greater impact on drug exposure than ECMO. However, larger prospective studies are required to confirm this observation.’ (lines 221 to 223 from the new manuscript).
- Interpretation of Findings - The finding that mechanical ventilation was an independent determinant of clearance is intriguing, but it could also reflect illness severity as a confounding factor. This point deserves a more nuanced discussion.
We are grateful for your observation. We have included the mean SOFA value of the patients in the Table 1 and there were no significant differences between both groups.
There are current studies showing modifications in PK/PDs’ of critically ill population leaded by mechanical ventilation as:
- DOI: 1007/s40262-023-01213-x
- DOI: 1055/s-0042-1744447
- DOI: 1097/MCC.0b013e3283374b1c
However, there are no specific studies relating the mechanical ventilation with the b-lactam PK/PD modifications, and this data should be taken more cautiously. We have modified the sentence in the discussion section:
‘MV has been reported to influence cardiac output as well as renal, hepatic and splanchnic perfusion, in addition to modifying the intrathoracic pressure, thereby leading to changes in PK parameters [45]. Although previous studies have suggested that MV may increase the volume of distribution and reduce the clearance of hydrophilic drugs, no statistically significant effects have been demonstrated for β-lactams [11,21].’ (lines 261 to 266 from the new manuscript).
- Methodological Considerations - The use of a fixed unbound fraction (fu) for TDM calculations may underestimate variability in critically ill patients, especially those with hypoalbuminemia. This limitation should be emphasized more clearly.
Thank you for your suggestion, we have added this as a limitation in the discussion section.
‘Free concentrations were calculated using fixed unbound fraction which could underestimate patients’ real Css, especially in those with hypoalbuminemia.’ (line 290 to 291 of the new manuscript).
- Methodological Considerations - Additionally, about half of the isolates lacked actual MIC data. Relying on surrogate MICs—particularly for piperacillin—may overstate the rate of PK/PD failure. This issue should be addressed in the discussion.
According to the reviewer’s suggestion we have addressed the issue in the discussion section.
‘The low attainment rates in our groups likely reflect the lack of microbiological isolation. The use of the theoretical MIC for piperacillin to assess PK/PD target attainment, may overestimate the actual MIC of Pseudomonas aeruginosa in clinical isolates and overstate the rate of target failure.’ (lines 253 to 257 of the new manuscript).
‘Moreover, the use of a surrogate MIC for piperacillin because only half of patients had a positive culture with a pathogen isolation, may have overestimated the PK/PD target failure rate, especially for microorganisms with lower MICs. Future studies incorporating organism-specific MICs in whole samples may provide a more accurate assessment of PK/PD attainment.’ (lines 285 to 289 of the new manuscript).
- Methodological Considerations - It would also be useful to specify whether all ECMO patients were on veno-venous or veno-arterial support, since ECMO mode can influence PK.
Following the reviewer’s suggestions, we added more information about ECMO mode, membrane and blood pump. We have included this in the methods section and as supplementary material.
‘Most patients (83.3%) received veno-venous ECMO support, 10 and 5 patients in the piperacillin and meropenem cohort respectively (Table S1).’ (lines 145 to 147 of the new manuscript).
‘Table S1. ECMO characteristics of the recruited patients’
|
Patient ID |
Drug |
Indication |
Duration |
ECMO mode |
Oxygenator membrane |
Bloodpump |
|
1 |
Piperacillin |
ARDS |
75 |
VV |
Maquet Rotaflow |
HLS |
|
8 |
Piperacillin |
ARDS |
123 |
VV |
Maquet Rotaflow |
PLS |
|
13 |
Piperacillin |
ARDS |
55 |
VV |
CardioHelp |
HLS |
|
18 |
Piperacillin |
ARDS |
25 |
VV |
Maquet Rotaflow |
PLS |
|
24 |
Meropenem |
ARDS |
21 |
VV |
CardioHelp |
HLS |
|
27 |
Piperacillin |
ARDS |
45 |
VV |
CardioHelp |
HLS |
|
35 |
Meropenem |
ARDS |
63 |
VV |
Centrimag |
HLS |
|
43 |
Piperacillin |
ARDS |
49 |
VV |
Maquet Rotaflow |
PLS |
|
48 |
Meropenem |
ARDS |
8 |
VV |
Maquet Rotaflow |
PLS |
|
52 |
Meropenem |
ARDS |
40 |
VV |
CardioHelp |
HLS |
|
55 |
Piperacillin |
ARDS |
40 |
VV |
CardioHelp |
HLS |
|
60 |
Meropenem |
ARDS |
43 |
VV |
Maquet Rotaflow |
PLS |
|
64 |
Piperacillin |
ARDS |
56 |
VV |
Maquet Rotaflow |
PLS |
|
67 |
Piperacillin |
ARDS |
29 |
VV |
CardioHelp |
HLS |
|
70 |
Piperacillin |
ARDS |
21 |
VV |
CardioHelp |
HLS |
|
72 |
Piperacillin |
ARDS |
23 |
VV |
CardioHelp |
HLS |
|
111 |
Meropenem |
Acute myocardial infarction |
11 |
VA |
CardioHelp |
HLS |
|
186 |
Meropenem |
Acute myocardial infarction |
13 |
VA |
CardioHelp |
HLS |
ARDS: Acute respiratory distress syndrome; VV: Veno-venous; VA: Veno-arterial.
- Methodological Considerations - Furthermore, the decision not to use mixed-effects modeling should be justified more clearly, as this may be questioned by readers familiar with pharmacometric methods.
Thank you for this comment. The mixed models’ analysis generally requires a substantial sample size in order to detect statistically significant effects within a complex experimental design, accounting for random effects and repeated measures. However, when research questions can be satisfactorily addressed by either a quantitative or qualitative analysis independently, a mixed-methods approach may not necessarily be the most appropriate option.
In this context, we did not apply mixed-effects modelling in our study for several reasons:
- The relatively small sample sizes of the subgroups.
- The presence of dichotomous categorical variables (ECMO, MV) as factors, without the nested subfactors (a feature often required in mixed models).
- The absence of markedly random variables in our subsamples.
The use of Generalised Estimating Equation (GENLIN), enabled us to incorporate repeated temporal variables while avoiding the added complexity—both in implementation and interpretation—associated with mixed-effects models.
We believe this approach provided a robust and transparent analysis, while remaining consistent with the characteristics of our data and the objectives of the study.
We have modified the methods section:
‘Mixed-effects models were not used due to the absence of scale variables.’ (lines 409 to 410 of the new manuscript)
- Results - The results are comprehensive and well-documented, but the tables are dense and somewhat difficult to interpret. Some could be simplified or moved to supplementary material. Presenting selected results graphically—for example, using boxplots or forest plots of fCss/MIC ratios—would make the data more accessible and highlight key differences more effectively.
As suggested, we have moved table 2 to supplementary table 1, presented results of fCss and fCss/MIC ratio achieved in a box plot (Figure 1) and modified the results section accordingly.
Results section:
‘Differences in clinical, analytical, infectious, and PK/PD parameters between the groups by sample are summarised in Table S2. Boxplots of fCss and fCss/MIC ratio are shown in Figure 1. Only 48.50% of patients had a positive culture with a pathogen isolation (63/130), and the actual MIC was only available in about 50% of those samples (Table S2).’ (lines 159 to 163 from the new manuscript).
Figure 1. Boxplot of meropenem and piperacillin (a) fCss and (b) fCss/MIC ratio sorted by group.
- (b)
The bottom and top extremes of the box represent the first (Q1) and the third quartile (Q3) range of the data, respectively (Q3–Q1: interquartile range). The dark horizontal line in the box is the median. ECMO = Extracorporeal membrane oxygenation. MIC = minimum inhibitory concentration. fCss = free plasma concentration at steady state. fCss/MIC = ratio of fCss to MIC.
- Discussion - The discussion integrates the findings with existing literature very well. However, it needs stronger critical reflection on possible confounding factors, such as the influence of the pandemic, differences in case severity, and selection bias.
Following the reviewer’s suggestion, we have acknowledged this issue as a limitation in the discussion section:
‘Other confounding factors related to severity of illness and type of patient not captured by the variables analysed may also have influenced the results obtained.’ (lines 282 to 283 from the new manuscript).
- Discussion - The conclusion that ECMO may not require specific dose adjustments should be softened. A safer interpretation would be that renal function and mechanical ventilation appear to play a greater role than ECMO in shaping drug exposure, but that larger prospective studies are needed for confirmation.
We appreciate the reviewer’s comment. Following this recommendation, we have replaced the sentence in the first paragraph of the discussion section to reflect a more tempered interpretation:
‘These findings suggest that renal function and MV appear to have a greater impact on drug exposure than ECMO. However, larger prospective studies are required to confirm this observation.’ (lines 221 to 223 from the new manuscript).
- Conclusion - The conclusion is concise and aligned with the study’s findings but should include a clearer cautionary statement about limited generalizability.
Following your suggestion, we have included a more cautionary statement about the limited sample size in the conclusion section.
‘Our results should be interpreted with caution due to the limited sample size and further prospective research is essential to validate these results in larger, more diverse populations and across varied geographical settings.’ (lines 418 to 420 from the new manuscript).
Comments on the Quality of English Language
Language & Style
Overall, the manuscript is readable, but the language would benefit from editing for clarity and conciseness. Several sentences are long and complex, which makes them harder to follow. Shortening these and eliminating redundancies would improve flow.
There are also occasional grammar issues (for example, “the throughout the dosing interval” should be “throughout the dosing interval”). Finally, word choices could be refined to maintain academic tone—for instance, “suffice” could be replaced with “may be adequate.”
We thank the reviewer for this valuable suggestion. Accordingly, we have revised the English to improve the readability.

Round 2
Reviewer 1 Report
Comments and Suggestions for Authors
the authors have revised along the suggestions provided, nothing to add